# Magnetometer Calibration and Field Mapping through Thin Plate Splines

**DOI:** 10.3390/s19020280

**Published:** 2019-01-11

**Authors:** Marco Muraccini, Anna Lisa Mangia, Maurizio Lannocca, Angelo Cappello

**Affiliations:** Department of Electrical, Electronic and Information Engineering, University of Bologna, Viale del Risorgimento, 2, 40136 Bologna, Italy; annalisa.mangia2@unibo.it (A.L.M.); maurizio.lannocca@unibo.it (M.L.); angelo.cappello@unibo.it (A.C.)

**Keywords:** movement analysis, wearable sensors, magnetometer calibration, Earth’s magnetic field mapping, thin plate splines

## Abstract

While the undisturbed Earth’s magnetic field represents a fundamental information source for orientation purposes, magnetic distortions have been mostly considered as a source of error. However, when distortions are temporally stable and spatially distinctive, they could provide a unique magnetic landscape that can be used in different applications, from indoor localization to sensor fusion algorithms for attitude estimation. The main purpose of this work, therefore, is to present a method to characterize the 3D magnetic vector in every point of the measurement volume. The possibility of describing the 3D magnetic field map through Thin Plate Splines (TPS) interpolation is investigated and demonstrated. An algorithm for the simultaneous estimation of the parameters related to magnetometer calibration and those describing the magnetic map, is proposed and tested on both simulated and real data. Results demonstrate that an accurate description of the local magnetic field using TPS interpolation is possible. The proposed procedure leads to errors in the estimation of the local magnetic direction with a standard deviation lower than 1 degree. Magnetometer calibration and magnetic field mapping could be integrated into different algorithms, for example to improve attitude estimation in highly distorted environments or as an aid to indoor localization.

## 1. Introduction

The Earth’s magnetic field has been used by the humans for centuries as a navigation tool. Evidence suggests that the animals use Earth’s magnetic field as well [1,2]. Moreover, some animals, such as spiny lobsters, are able not only to detect the direction of Earth’s magnetic field, they can even sense their true position relative to their destination [3]. Their ability seems to suggest that the knowledge of magnetic information could lead to the determination of the position.

It is well known that the Earth’s magnetic field is not a predefined and time-constant vector [4]. It depends on the latitude and longitude coordinates and suffers from different types of fluctuations due to the diurnal cycle, movement of magnetic poles, and more randomly, to geomagnetic storms caused by solar flares [5], but for all practical purposes, especially for the use of magnetic data in sensor fusion algorithms to estimate the local orientation of a Magnetic and Inertial Measurement Unit (MIMU), these fluctuations are negligible when compared to the typical point to point variation of magnetic field inside a building [4,6,7]. The local magnetic distortion depends on the position and is not known a priori. In fact, its intensity and direction strongly depend on the proximity of metallic objects with high relative permeability, such as iron reinforcements in buildings, permanent magnets, motors and electronic devices.

Until a few years ago, these distortions have been mostly considered as a source of error for compasses and magnetometers in indoor localization and in sensor fusion algorithms, as they interfere with compass direction. However, if the distortions are temporally stable and spatially distinctive [4,5,8], they provide a unique magnetic landscape that can be used for different purposes. With regard to this, Haverinen et al. [3] have shown that the anomalies in indoor magnetic fields can be manually collected to build a map that can be used to localize both robots and humans equipped with wearable sensors.

It should be noted that magnetic distortions are generated not only by external magnetic materials (e.g., iron reinforcements in buildings or electronic devices). In fact, there are also self-induced distortions caused by the sensing element itself [9,10]. These types of distortion could be divided into two groups: hard and soft iron [9]. These two types of effects arise, respectively, from permanent magnets and DC currents on the compass platform and from the interaction of the Earth’s magnetic field and any high permeability material on the same platform. While hard iron effects will remain constant in the MIMU reference frame for all compass orientations, soft iron effects do not remain constant as they vary with the orientation of the sensor relative to the direction of the Earth’s magnetic field. To compensate for these sources of distortion, many algorithms for calibrating the magnetometer data have been proposed [9,10,11]. All of them require an acquisition where the magnetometer explores all possible orientations.

In the literature different algorithms to interpolate and extrapolate a mapping of the magnetic field in an internal environment have been suggested [3,4,5,12,13]. Solin et al. [12] proposed a Bayesian non-parametric probabilistic modelling approach for interpolation and extrapolation of the magnetic field. In [4] the map is collected by a robotic platform with minimal sensor equipment. It is demonstrated that a simple magnetometer along with some odometric information suffices to construct the map via a Simultaneous Localization and Mapping (SLAM) procedure based on the Rao-Blackwellized particle filter to provide recursive Bayesian estimation.

To the best knowledge of the authors, all previous works use pre-calibrated data from the magnetometer to estimate the magnetic field map in the measurement volume. This implies that at least two acquisitions must be performed in order to: (i) calibrate the magnetometer, and (ii) estimate the magnetic field map. In this paper a new technique is proposed, based on a single acquisition, for the simultaneous: (i) calibration of the magnetometer, and (ii) estimation of the Earth’s magnetic field map in a fixed system of reference. The approach relies on the use of 3D-TPS, a spline-based technique for data interpolation and smoothing [14].

Reviewing the relevant literature, it appears that most of the work related to the estimation of magnetic map is focused on 2D mapping [3,4,5,13,15], even if a 3D approach is certainly preferable in applications such as robotics, biomechanics and drone control.

The information obtained could be integrated into sensor-fusion algorithms for the orientation estimate. Several studies have highlighted the lack of accuracy in the orientation data provided by MIMU in the movement analysis laboratory due to the presence of irregularities caused by iron reinforcements in floors, walls and ceilings, or other equipment [7,16,17]. Currently, there is no robust solution in the literature regarding this issue.

Different strategies have been adopted to tackle this problem [6,18]. When distortions have short duration a possible solution is to complete the Kalman Filter dynamical model with additional equations modelling the magnetic drift using random-walk or first-order Gauss-Markov models [18].

When the magnetometer output is corrupted by the presence of electromagnetic devices over a long period of time this approach does not work properly. The knowledge of the field map could make it possible to overcome these limitations.

Another area where these maps could be used is for indoor localization purposes, thereby offering a promising alternative to traditional methods for achieving GPS-level localization indoors [15]. Traditionally, Simultaneous Localization And Mapping (SLAM) in robotics has been tackled with laser or vision-based approaches [8]. However, other more unconventional sensor approaches for SLAM have recently attracted attention, such as WiFi-based methods [19,20,21] or depth sensors [22]. The use of magnetometer for SLAM represents a new and low-cost challenge [8].

The aim of this study is to provide and test a single procedure to simultaneously calibrate the triaxial magnetometer and map the 3D magnetic field in the acquisition volume. To this end, a model based on thin plate splines (TPS) interpolation was developed and tested using both simulated and experimental data. The results suggest that it is possible to obtain an accurate and stable 3D description of the magnetic field in the measurement volume.

## 2. Materials and Methods

### 2.1. Model of a Triaxial Magnetometer in the Presence of Magnetic Field Distortion

Two right-handed orthogonal reference systems are defined: A Laboratory Reference System (LRS), which is assumed to be fixed to the Earth and a Sensor Reference System (SRS) that is integrated with the magnetometer.

In this work, the axes of LRS are defined by reference to the Nord-West-Up (NWU) convention in which the *x*-, *y*- and *z*-axes point to magnetic north, magnetic west and upward respectively [6]. The above-mentioned convention makes it possible to build an LRS without any external reference, except for gravity and magnetic field.

The magnetic field at a given point P(*x*, *y*, *z*) is given by the contribution of both the Earth’s magnetic field and the local distortion. In a short observation time interval, it can be assumed that the Earth’s magnetic field is a constant vector, $B_{0}$ (Gauss), in the whole measurement volume. Furthermore, let us assume that the local distortion at point P, ΔB(P) is time-constant only in presence of motionless high permeability materials or static magnetic field generators in or near the measurement volume. In the proposed algorithm, and consequently in the following equations, local distortions were assumed temporally stable and spatially distinctive.

An ideal triaxial magnetometer measures the components of the local magnetic field along the three mutually orthogonal axes, *x*, *y*, *z*, of the SRS. The ideal magnetometer output at time *k* (called also frame) is given by Equation (1):(1)$$m_{k}^{I}=R_{k}^{T}\;\left[B_{0}+\Delta B\left(P_{k}\right)\right]=R_{k}^{T}\;B\left(P_{k}\right)$$ where $R_{k}$ is the time-varying orientation matrix of the SRS with respect to the LRS, and $P_{k}$ is the position vector of the sensor.

Under actual conditions, we must take into account instrumentation errors and magnetic deviation. The instrumentation errors include: (i) cross-talk effects due to non-orthogonal measurement axes, (ii) non-unitary and generally different gains along the three axes, (iii) offset and iv) additive noise, supposed as a Gaussian wideband noise [10] as reported in [23]. The magnetic deviation includes: (v) soft iron and (vi) hard iron effects. Both instrumentation errors and magnetic deviation can be accounted for by using a simple linearized model [10,24], expressed at frame *k*:(2)$$m_{k}=Wm_{k}^{I}+O+v_{k}$$ where $m_{k}$ is the actual measured magnetic field, W a 3 × 3 matrix depending on (i), (ii) (v), O a 3 × 1 bias vector depending on (iii) and (vi), and $v_{k}$ the additive noise.

For magnetometer calibration, W and O must be estimated. Several studies have proposed different methods for estimating optimal calibration parameters [9,10,11].

The first innovative contribution of this work is to model the local magnetic field map by using a 3D Thin Plate Spline (TPS) interpolant [14]. TPS are a spline-based technique for data interpolation and smoothing. Its name refers to a physical analogy with the bending of a thin sheet of metal. Just as the metal has rigidity, so the TPS fit also resists bending, implying a handicap involving the smoothness of the fitted surface. Because of its elegant algebra expressing the dependence of the physical bending energy of a thin metal plate on point constraints, the TPS represents a valid tool for interpolating surfaces over scattered data.

Since acquiring the true field is infeasible, it is approximated by an estimate based on a finite number of data points. The estimate can be modelled in a multitude of ways; in this work a TPS was selected. All the formulations presented in this work follow the definitions in [14].

Using a 3D-TPS interpolant the local magnetic field can be expressed in the LRS by this equation:(3)$$B\left(P\right)=B_{w}+KP+\sum_{i=1}^{nk}V_{i}f_{i}\left(P\right)\;\mathrm{where}\;f_{i}=\|P-P_{i}\|$$ where $B_{w}$ is a 3 × 1 vector expressing the constant magnetic field in LSR, K is a 3 × 3 matrix which expresses a linear relationship between P and B(P), nk is the number of kernel points, $f_{i}$ is a scalar value expressing the distance between a generic point P and the *i*-th kernel point $P_{i}$ and $V_{i}$ is a 3 × 1 vector. It can be observed that, in absence of magnetic distortion, the term $B_{w}$ coincides with the undisturbed Earth magnetic field, expressed in LRS.

In the next sections, regarding to Equation (3), nk, P and $P_{i}$ will be assumed known, while $B_{w}$, K, and $V_{i}$, *i* = 1, nk will be the unknown parameters to be estimated for obtaining the magnetic field map. The number and the position of the kernel points should be chosen by considering the distortion properties. The spatial sampling interval should be reduced where the magnetic field gradient is higher.

Equations (1)–(3) give the final model, in the *k*-th frame:(4)$$m_{k}=WR_{k}^{T}\left[B_{w}+KP_{k}+\sum_{i=1}^{nk}V_{i}f_{i}\left(P_{k}\right)\right]+O+v_{k}$$

Referring to (4), $m_{k}$, $P_{k}$ and $R_{k}$ will be the input of the proposed algorithm. Specifically, $m_{k}$ will be the magnetometer output, while $R_{k}$ and $P_{k}$, expressing the orientation and the position of the SRS with respect to the LRS, will be provided by an external measuring system. In this work, this is an optical motion capture system.

On the contrary, all the parameters related to the description of the magnetic distortion (W, $B_{w}$, K, $V_{i}$, O) will be estimated.

### 2.2. Parameter Estimation

The second original contribution of this paper is the simultaneous estimation of the parameters related to the magnetometer calibration (W, O) and to the TPS model ($B_{w}$, K, $V_{i}$).

Since Equation (4) is still valid if the first element of W ($w_{11}$) is multiplied and B(P) is divided by the same scalar value, $w_{11}$ has been assumed equal to 1. Applying this transformation, the number of parameters to be estimated is 23 + 3 nk.

Equation (4) can be re-written as follows:(5)$$m_{k}=WR_{k}^{T}M_{k}\theta +O+v_{k}=A_{k}\theta +O+v_{k}$$ where θ is a (12 + 3 nk) × 1 vector containing all the parameters related to $B_{w}$, K and $V_{i}$. $M_{k}$ is a 3 × (12 + 3 nk) matrix defined according to Equation (4). In particular, $M_{k}$, θ and $A_{k}$ are defined as follows:(6)$$M_{k}=\left[\begin{array}{cccccccccccccccccccccc} & x_{k} & y_{k} & z_{k} & 0 & 0 & 0 & 0 & 0 & 0 & f_{1} & f_{2} & & f_{nk} & 0 & 0 & & 0 & 0 & 0 & & 0\\ I_{2} & 0 & 0 & 0 & x_{k} & y_{k} & z_{k} & 0 & 0 & 0 & 0 & 0 & \ldots & 0 & f_{1} & f_{2} & \ldots & f_{nk} & 0 & 0 & \ldots & 0\\ & 0 & 0 & 0 & 0 & 0 & 0 & x_{k} & y_{k} & z_{k} & 0 & 0 & & 0 & 0 & 0 & & 0 & f_{1} & f_{2} & & f_{nk} \end{array}\right],\;A_{k}=WR_{k}^{T}M_{k}$$
$$\theta =\left[\begin{array}{c} B_{w}\\ K{\left(1,:\right)}^{T}\\ K{\left(2,:\right)}^{T}\\ K{\left(3,:\right)}^{T}\\ V_{1}\\ V_{2}\\ .\\ .\\ .\\ V_{nk} \end{array}\right]$$

It should be noted that, once $P_{k}$ and $P_{i}$ are defined, $M_{k}$ is fully determined. So, referring to Equation (5), the magnetometer output, once W defined, is linearly related to the unknown parameters θ and O. The least squares solution can be easily found by the pseudoinverse method [25].

The parameter estimation problem is solved using an iterative method according to the following steps:(1)Initialization: $P_{w}=$ [$w_{12\;}\;w_{13\;}w_{21}\;w_{22\;}\;w_{23\;}\;w_{31}\;w_{32\;}w_{33}$]’; where $\;w_{ij}$ represent the elements of the W matrix and $w_{11}$ = 1(2)$A_{k}=WR_{k}^{T}M_{k}$(3)Linear least squares estimation of θ and **O** through Equation (5)(4)Calculation of the 3*n* × 8 sensitivity matrix **S** of the output residual vector $V=\left[v_{1}\;;\;v_{2}\;;\;\ldots ;v_{n}\right]$ with respect to $P_{\;w}$(5)The optimal correction is calculated as $\Delta P_{w}=-\mathrm{pinv}\left(S\right)$
**V**(6)Update $P_{w}$ ($P_{w}=P_{w}+\;\Delta P_{w}$)(7)Return to (2) until the cost function $V^{T}V$ is flat or below a given threshold.

### 2.3. Simulated Experiment

The first validation of the proposed model consists in a simulation study. This will demonstrate the validity of the proposed method under controlled data.

Referring to Equation (5), the magnetometer output was simulated, at each frame *k*, as a function of predefined parameters (W, O, $M_{k}$, θ), position ($P_{k}$) and orientation ($R_{k}$). Specifically, for each test, the following parameters were defined:(1)Laboratory Reference System, LRS(2)Sensor Reference System, SRS. The SRS defined by the cluster of markers is assumed coincident with the magnetometer SRS(3)The constant Earth’s magnetic field expressed in LRS;(4)Simulated calibration parameters (W ,  O) to apply to the magnetometer output(5)The number and positioning of kernel points(6)A set of parameters to identify the magnetic field map (M,  θ)(7)The trajectory, in term of position and orientation, of the SRS with respect to LRS

The data were simulated with a sample frequency of 100 Hz. For each frame, according to Equation (5) and the defined parameters, the simulated magnetometer output was calculated. In the simulated data W and O were randomly chosen in a way that their values do not deviate too much from the values found in real acquisition. In particular, in each simulated test, a vector containing 12 elements was generated with a uniformly distributed random sequence in the interval (−0.2, 0.2). In order to create realistic calibration parameters, this vector was added to the ideal calibration parameters, corresponding to the identity matrix for W and null vector for O.

A gaussian noise, with zero mean and predefined standard deviation was added, for each simulated test, to the magnetometer output. In this way, the effect of the measurement noise on the parameter estimation was evaluated. The range of noise standard deviation varies from 0.0015 G to 0.01 G, according to magnetometer specifications [23]. This range of standard deviations was chosen to simulate noise behavior in real data acquisition. More specifically, 0.0015 Gauss corresponds to the noise standard deviation found on the real magnetometer [23] after filtering the signal with a second order bidirectional low-pass filter with cut-off frequency 5 Hz, while 0.01 Gauss represents the worst condition.

The kernel points used to model the magnetic field were uniformly distributed along the 3 axes of the laboratory reference system (LRS). Two different classes of tests were conducted.

#### 2.3.1. Class of Tests 1

In the first class of tests, random rotations of the SRS with respect to the LRS exploring all possible orientations are simulated in the 3D space while the sensor is moving all around in the acquisition volume. The trajectory of the magnetometer was randomly generated in a measurement volume of 1.5 × 1.5 × 1.5 m^3^.

#### 2.3.2. Class of Tests 2

In the second class of tests ‘incomplete’ data acquisition is simulated. Several studies [9,10,11] have reported the need to explore every rotation while the magnetometer calibration procedure is performed. Otherwise, this could lead to an inaccurate calibration. Therefore, in these tests, the magnetometer explores different positions in the measurement volume (the same volume of the class of tests 1), but without exploring all the rotations. Specifically, the rotations were generated with a uniformly distributed random sequence in the intervals (0, π/5), (0, π/6), (0, π/3) around *x*-axis, *y*-axis, *z*-axis, respectively, of the local reference system.

### 2.4. Real Data Acquisition

To validate the proposed approach using real data, a set of experiments was performed. All the data were collected in the Movement Analysis Laboratory at the University Sports Center “Record” in Bologna. An optical motion capture system (BTS SMART-DX 7000) with 10 cameras was used as a reference to obtain the position and orientation of the SRS with respect to LRS. A force plate integrated in the motion capture system was used to synchronize stereo photogrammetric data with MIMU data. Two wireless MIMUs were used to collect inertial and magnetic data. The MIMUs were developed by NCS Lab (Carpi, Italy), integrating an accelerometer, a gyroscope and a magnetometer all on a single board. The magnetometer’s datasheet can be found in [23]. The entire acquisition lasted from 2 to 3 min. In order to create an inhomogeneous magnetic field, the environment was filled with ferromagnetic objects, as shown in the Figure 1.

MIMU outputs were sampled at 120 Hz, while marker positions were estimated by stereo-photogrammetry at a 250 Hz frame rate. The MIMU was fixed on a stick and moved within the measurement volume. Both at the beginning and at the end of the acquisitions, the force plate was hit with the stick to synchronize the two signals, using peaks revealed on both force plate and accelerometer. After the synchronization, the stereo-photogrammetric data were down-sampled at 120 Hz. Both magnetometer data and stereo-photogrammetry data were digitally filtered through a second order bidirectional Butterworth low-pass filter with 5 Hz cut-off frequency. As for the simulated data in class of tests 1, the MIMU in the real acquisition explored the whole measurement volume (3 × 1.5 × 1.3 m^3^) assuming all possible orientations. Five different acquisitions were performed, with two different MIMUs, with 2 h interval between the two MIMUs acquisitions. In this way, it was possible to assess algorithm performance using different MIMUs and to test the magnetic distortion temporal stability. In all the analyses performed for the reconstruction of the magnetic field, kernel points were uniformly distributed along each axis of the laboratory reference system (LRS).

### 2.5. Output Evaluation

The performance of the proposed method was evaluated in different ways, as illustrated below.

#### 2.5.1. Intra-Dataset Magnetic Field Reconstruction Errors

For both simulated and real tests, after the parameter estimation, the reconstructed magnetometer output, corresponding to the right member of Equation (5), and depending on the estimated parameters, was calculated for each frame *k*. In this way, a comparison of the real magnetometer output ($m_{k}$) with the reconstructed magnetometer output is possible along the three axes. Root Mean Square Errors (RMSE) along the three axes were calculated. In order to quantify the effect of the magnetic distortion, the angle between the *x*-axis of the LRS (NWU convention) and the magnetic north is also computed. If no distortion is present, this angle is equal to 0 degrees, otherwise the errors in the estimation of this angle will be called ‘heading errors’, with a clear reference to the so-called heading angle, as defined in [9]. With regards to the simulated data, for each frame, the angle between the vector pointing to the magnetic north and the *x*-axis of the LRS is known for construction. Because of this, the heading error is simply calculated as the difference between the true and estimated angle. In real data, a gold standard is not available, and the heading error is defined as the difference between the angle computed from real magnetometer measure, and the angle computed from reconstructed magnetometer measure (right member of Equation (5)). Both for simulated and real data, heading angle error was calculated in terms of RMSE and indicated as RMShE.

#### 2.5.2. Influence of Number of Kernel Points and Training Group Dimension

Each dataset was modelled using different kernel configurations; in this way, the influence of the number and position of kernel points was evaluated. Moreover, a training-testing procedure was implemented with the aim of investigating the number of frames necessary for the parameter estimation. In this procedure, the algorithm was trained on a predefined number of acquisition frames and the parameters estimated in the training session were tested on the remaining part of the data. The errors mentioned above were evaluated both as a function of the number of kernel points used in the TPS model and as a function of the number of frames used for training the algorithm.

#### 2.5.3. Magnetic Field Comparison Over Different Datasets

Finally, in real data acquisition, a comparison of the magnetic field estimated in different dataset was performed. Let us assume that $B_{1}\left(P\right)$ and $B_{2}\left(P\right)$ are the magnetic field expressed in the LRS at point P estimated in two separate trials. In order to compare these two different results, a dataset of $P_{k}$ was artificially created. For each $P_{k}$, $B_{1}\left(P_{k}\right)$ and $B_{2}\;\left(P_{k}\right)$ were calculated. Both $B_{1}\left(P_{k}\right)$ and $B_{2}\left(P_{k}\right)$ are 3 × 1 vector indicating the magnetic field estimated in trial 1 and 2, respectively, at $P_{k}$. In this way, for each frame *k*, a comparison of $B_{1}\left(P_{k}\right)$ with $B_{2}\left(P_{k}\right)$ is possible. In sensor-fusion algorithms for the attitude estimation using MIMUs, it is essential to know the angle on the horizontal plane between magnetic north and the *x*-axis of the LRS (declination angle). Because of this, this angle was calculated for each frame *k*, both for $B_{1}\left(P_{k}\right)$ and $B_{2}\left(P_{k}\right)$. Errors were expressed as RMSE. The results emerging from the comparison of the magnetic field estimated in different dataset have multiple meanings. First, it allows one to analyze the temporal stability of the magnetic distortion inside the building. In fact, the different dataset used to construct to magnetic map were not acquired at the same time. The finding that the reconstructed magnetic map is very similar in all the trials could be an important point for assessing the temporal stability of magnetic distortion in the measurement volume. Then, it offers the possibility to analyze the optimal number of kernel points for the description of the field. In fact, a large number of kernel points may not be the best option. Using a very high kernel number may produce very low intra-test errors, but the ability to generalize is lost.

## 3. Results

### 3.1. Simulated Experiment

In this subsection the results emerging from simulated experiment will be presented, both from class of tests 1 and 2.

#### 3.1.1. Intra-Dataset Magnetic Field Reconstruction Errors

A comparison between the reconstructed magnetometer output and the simulated magnetometer output is presented graphically and numerically. In all the analyses conducted in this paragraph, 27 uniformly distributed kernel points were used to model the field. Results emerging from class of tests 1 are reported in Figure 2, in terms of both errors in the magnetometer reconstruction and heading errors. The errors in the estimation of the magnetometer output and the heading angle were reported in Table 1 in terms of RMSE.

For both classes of tests, the mean error in heading estimation is almost zero. In this case, the RMSE is very close to the error standard deviation.

#### 3.1.2. Influence of Number of Kernel Points and Training Group Dimension

The heading errors, in terms of RMSE, were evaluated as a function of both the number of kernel points used in the TPS model (Figure 3a) and of the number of frames used for training the algorithm (Figure 3b).

### 3.2. Real Data Acquisitions

In this subsection the results emerging from real data acquisitions will be presented.

#### 3.2.1. Intra-Dataset Magnetic Field Reconstruction Errors

The analysis adopted in simulated data were replicated on real data. In Figure 4a,b are reported, respectively, an example of a comparison between reconstructed magnetometer output with the real magnetometer output and the related heading error.

Table 2 summarizes the errors in heading estimation for each data acquisition, as a function of the number of kernel points used to interpolate the magnetic field map. 

#### 3.2.2. Influence of Number of Kernel Points and Training Group Dimension

Heading errors were evaluated as a function of both the kernel number used in the TPS model in a single acquisition and the different percentages of the frames used for model training. The heading RMSE is reported as a function of kernel number in Figure 5a and as a function of the percentage of frames in Figure 5b.

#### 3.2.3. Magnetic Field Comparison Over Different Datasets

Finally, a comparison of the LRS magnetic field estimated in different acquisitions was performed. An example is reported in Figure 6, where two different LRS magnetic fields estimated in two different acquisitions, with different MIMUs, are compared:

Figure 6 highlights that the use of many kernel points leads to an over-fitting of the field with consequent reduced generalization ability.

## 4. Discussion

This paper suggests a novel method based on TPS for mapping 3D magnetic distortion in an acquisition volume. Moreover, the algorithm enables the simultaneous estimation of the magnetometer calibration parameters.

Mapping the distortion of the field to obtain its component expressed in an LSR in every point of the acquisition volume could be useful in many applications, like attitude estimation in sensor-fusion algorithms [18] and indoor localization [26].

Several authors [3,5,8,12,15] proposed different methods for the environment magnetic map reconstruction. All these works are based on the use of pre-calibrated magnetometer data. This implies that at least two acquisitions must be done: one for calibrating the magnetometer data and the other for mapping the distortion in the environment. The present work overcomes this limitation by demonstrating that the simultaneous estimation of both magnetometer calibration parameters and 3D magnetic mapping is possible.

The validity of the proposed method was demonstrated using both simulated and real data. More specifically, using simulated data, it was possible to gain a full knowledge of the noise statistical properties and to test many different parameter configurations related to the magnetometer calibration and magnetic field map. In this way it was possible to test the robustness of the algorithm in the parameters estimation in different environments only by changing the simulated magnetic distortion.

The results emerging from simulated experiments confirm that, under predefined conditions regarding data acquisition, all parameters are uniquely identifiable. In particular, if the magnetometer explores every possible orientation while it is moving in a random way in the measurement volume, all the parameters related to the calibration (**W**, **O**) are estimated with negligible errors. This implies that the reconstructed magnetometer output is almost identical to the simulated data (Table 1). The error between them is Gaussian with null mean and standard deviation equal to the standard deviation of the noise added to the simulated data (Table 1), for all the configurations tested. Even the heading angle was estimated with excellent results (Table 1), with a standard deviation error of about 1 degree. All these results confirm the possibility to estimate, in a single procedure, all the parameters of the model shown in Equations (4) and (5).

The importance of exploring all the possible rotations to obtain a good accuracy in the magnetometer calibration parameters has been underlined by several authors [9,10] and it was confirmed by class of tests 2, where ‘incomplete’ data were simulated. Comparing the results of class of tests 1 with class of tests 2, higher errors emerged in the second tests for both calibration parameters and heading angle estimation (Table 1).

In Figure 3a the heading error is shown as a function of the number of kernel points used to describe the map. In particular, as expected, the Root Mean Square heading error decreases as the number of kernel points increases. Therefore, the simulated tests showed that the errors depend on different causes: (i) how the acquisitions were made, (ii) the noise added to the simulated signal, (iii) the number (and positioning) of kernel points used to describe the map.

In real experiments these findings were confirmed. The parameters estimation of both magnetometer calibration and magnetic field mapping in real data acquisition leads to an excellent and stable reconstruction of indoor magnetic field, with errors in the heading estimate close to 1° (Table 2). This result is comparable with the one reported in [9] where it is shown that the post-calibration residuals result in a system with heading errors in the order of 1 to 2 degrees. Comparing the accuracy reached for mapping the magnetic field with similar papers [27,12], the results obtained are fully satisfactory. For example, in [27] it was shown that the mean heading error, after magnetic field mapping, was around 2 degrees.

The comparison of the results obtained by real and simulated data is encouraging. In fact, comparing Figure 2 and Figure 4, the magnetic field reconstruction errors show the same qualitative behavior. Errors found in real data are greater than the ones found in simulated data (Figure 2b, Figure 4b). This can be explained by different factors emerging in real data acquisitions, including: *(i)* not perfect linearity of the magnetometer output with respect to the calibration parameters (referring to Equation (2)); *(ii)* errors in the reconstruction of position and orientation from the data acquired by the stereo-photogrammetric system.

The temporal stability of the magnetic distortion was proved by comparing the results obtained in different acquisitions performed two hours apart (Figure 6). This result is an important confirmation of the stability of magnetic field in buildings [5].

Regarding the error in the estimation of the heading angle as a function of the number of kernels, two different behaviours have been noticed. The first one is clearly demonstrated in Figure 5a, where the RMShE is reported as a function of the kernel points number, within the same acquisition. It is shown that, with the use of a high kernel points number, the errors decrease. The second one is nicely expressed in Figure 6, in which a comparison of the heading angles estimated in two different acquisition is performed. This figure clearly illustrates that there is an optimal kernel configuration to guarantee optimum mapping. More specifically, while Figure 5a illustrates the intra-dataset magnetic field reconstruction errors, Figure 6 shows the ability to generalize the results obtained in a dataset over different datasets.

Heading errors were evaluated also as a function of the different percentages of the frame used for the training model, as reported in Figure 3b for simulated data and in Figure 5b for real data. A training-testing procedure was implemented to understand the predictive capabilities of the model. To this end, the model was trained on a predefined number of frames, and the parameter estimated were tested on the remaining frames. Results indicate that is not necessary to use every available frame in the analysis, because, as it can be seen in Figure 5, the error remains constant for percentages greater than 10%. This percentage depends both on the distortion properties and on the way in which the data are collected.

One limitation of this study may be the acquisition volume size. All the real data were acquired in a volume of 3 × 1.5 × 1.3 m^3^. Further studies may include a bigger acquisition volume. In that case, the number of kernel points should probably be increased to achieve the same degree of accuracy found in this work. Another point to underline is that the proposed algorithm is based on the assumption of a time-invariant magnetic field (3) during data acquisition. If the hypothesis is not verified, Equations (3) and (4) should include terms that vary with time in a known way.

Future developments of the present work will address the study of the number and positioning of kernel points, in relation to the magnetic distortions present in the acquisition volume. When the field is almost constant, no kernel points are required to model it. Likewise, in the presence of high magnetic distortion, a greater number of kernel points will be needed. Furthermore, within the measurement volume, some regions could be heterogeneous and others could be homogeneous. In this case it would be appropriate to choose not only the number of kernels, but also their position. In fact, the number and position of the kernel points should be chosen by taking into account the distortion properties. The spatial sampling interval should be reduced where the magnetic field gradient is higher. Automated algorithm to find the optimal number and positioning of kernel points might be the subject of a future study.

## Figures and Tables

**Figure 1 sensors-19-00280-f001:**
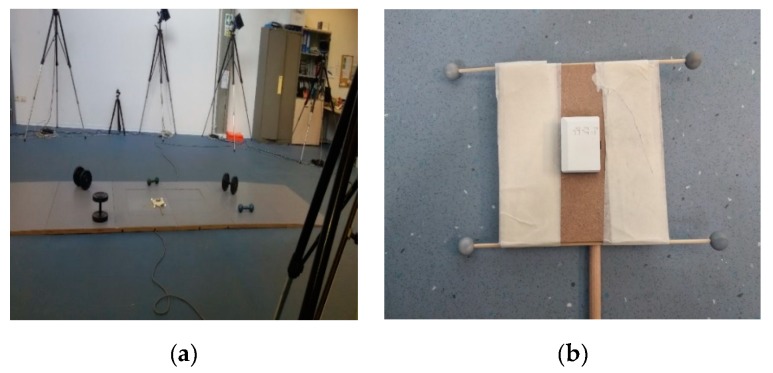
(**a**) The figure illustrates the acquisition volume. A force plate was positioned in the center of the volume, surrounded by different ferromagnetic objects. The acquisition volume was approximately 3 × 1.5 × 1.3 m^3^; (**b**) cluster of four markers used for the acquisition by stereo-photogrammetry. The inertial and magnetic sensor was fixed to the cluster frame and axis alignment was done manually. All residual misalignments were included in the W matrix.

**Figure 2 sensors-19-00280-f002:**
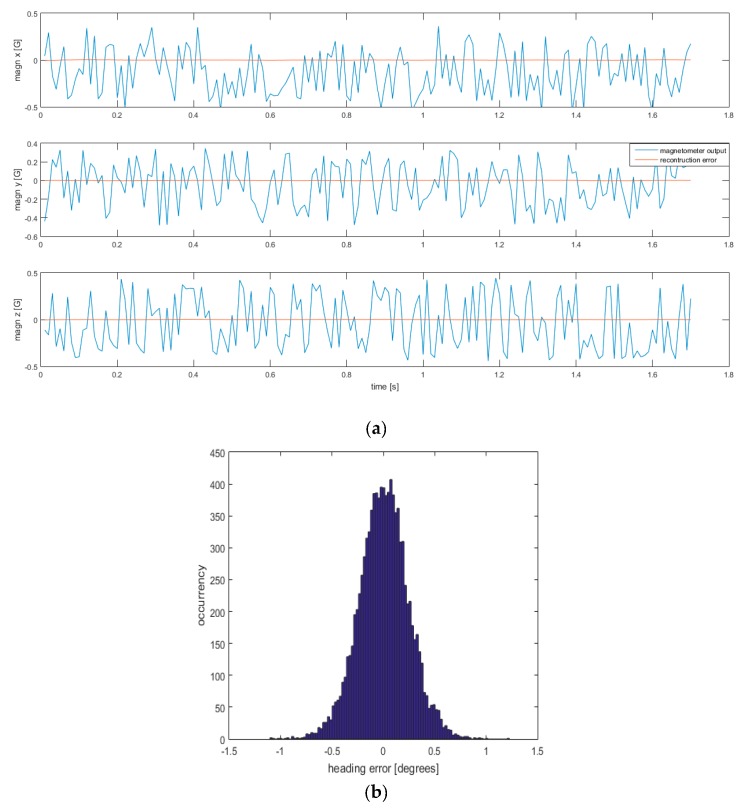
(**a**) In Figure 2a the simulated magnetometer outputs and the residuals are plotted in blue and red respectively along *x, y, z*-axes. The error has the same mean and standard deviation along all the simulated experiment (120 s) and therefore only the first 1.7 s are reported here; (**b**) histogram of the heading error.

**Figure 3 sensors-19-00280-f003:**
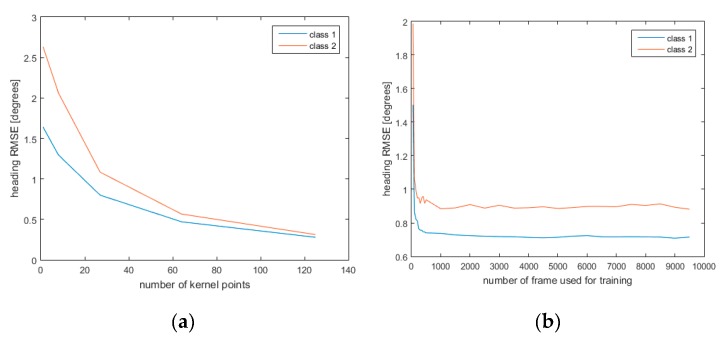
(**a**) Heading RMSE as a function of the number of kernel points used to interpolate the data in the TPS model. The number of frames used for training the algorithm was 7000. (**b**) Heading RMSE as a function of the number of frames used for training the algorithm. The number of kernel points was fixed at 27.

**Figure 4 sensors-19-00280-f004:**
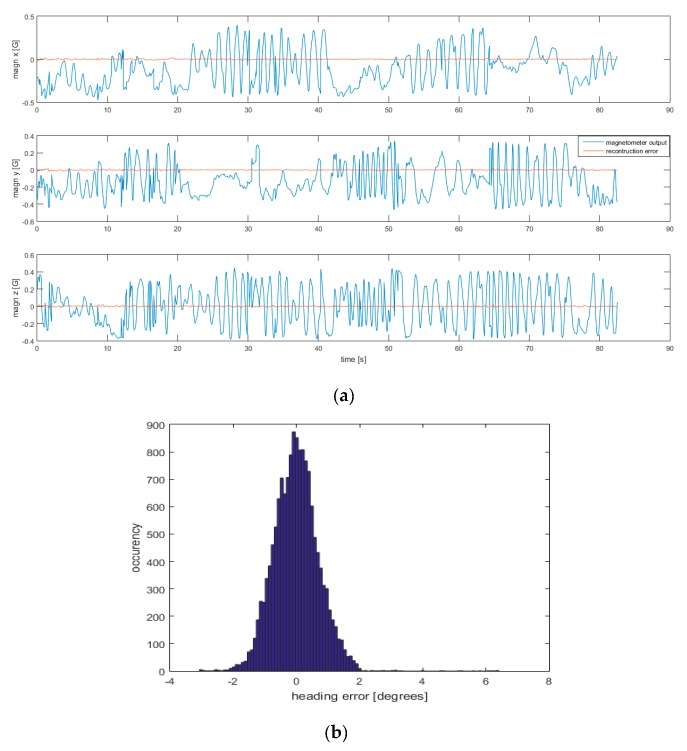
(**a**) example of magnetometer outputs (blue line) and reconstruction errors (red line) along the three local axes; (**b**) histogram of the heading error. The number of kernel points was fixed at 27.

**Figure 5 sensors-19-00280-f005:**
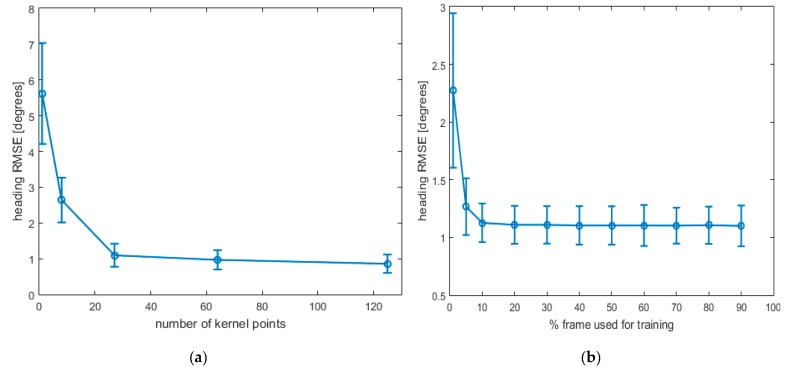
(**a**) Heading RMSE as a function of kernel points number. The percentage of the frames used for the model training was 75%; (**b**) heading RMSE as a function of the percentage of frames used for the model training. The number of kernel point was fixed at 27. The observation interval in all experimental sessions varies from a minimum of 14,050 to a maximum of 20,460 frames, corresponding 120–180 s at 120 Hz sampling frequency. In Figure 5a,b means ± standard deviations over all the real data acquisitions were reported.

**Figure 6 sensors-19-00280-f006:**
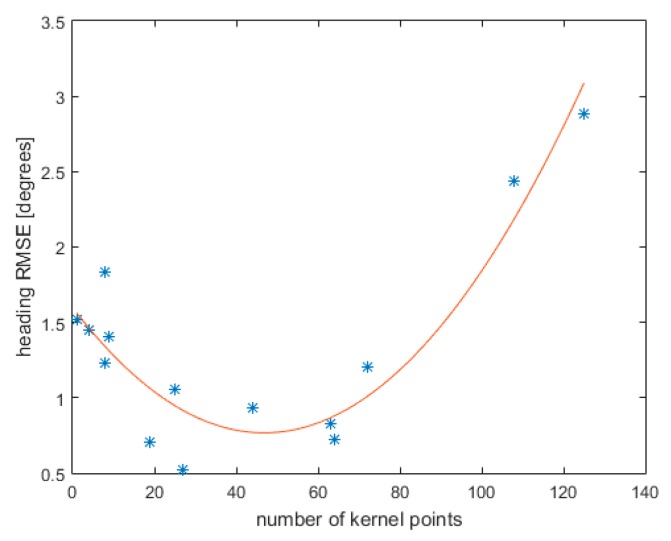
Heading angle RMSE between two different acquisitions as a function of the number of kernel points used in the TPS model. * represent the values directly obtained and the curve is a best fit parabola obtained by a least-squares method.

**Table 1 sensors-19-00280-t001:** Errors along the three magnetometer axes and heading errors in class of tests 1 and 2, respectively. The errors were reported in terms of RMSE as a function of the standard deviation of the Gaussian noise added to the simulated signals. Noise std indicates the standard deviation added to the simulated magnetometer data. RMShE indicates Root Mean Square heading Error. RMSE magn *x*, RMSE magn *y*, RMSE magn *z*, indicate the Root Mean Square Error along *x, y, z* axes, respectively, of the SRS.

**Class of tests 1**				
	**RMShE**	**RMSE magn *x***	**RMSE magn *y***	**RMSE magn *z***
Noise std = 0.0013 G	0.243°	0.0015 G	0.0015 G	0.0015 G
Noise std = 0.005 G	0.797°	0.005 G	0.005 G	0.005 G
Noise std = 0.01 G	1.595°	0.01 G	0.01 G	0.01 G
**Class of tests 2**				
	**RMShE**	**RMSE magn *x***	**RMSE magn *y***	**RMSE magn *z***
Noise std = 0.0013 G	0.339°	0.0022 G	0.0021 G	0.0023 G
Noise std = 0.005 G	1.092°	0.0051 G	0.0050 G	0.0053 G
Noise std = 0.01 G	2.178°	0.011 G	0.008 G	0.01 G

**Table 2 sensors-19-00280-t002:** The table summarizes the errors in heading estimation for each data acquisition as a function of the number of kernel points used to interpolate the magnetic field map. The errors are expressed in term of Root Mean Square heading Errors. The first row indicates the number of kernel points used for the interpolation. Mean errors ± standard deviation are reported in the final row.

	nk=8	nk=27	nk=64	nk=125
RMSE_MIMU1 test1 (°)	2.1	1.5	1.43	1.34
RMSE_MIMU1 test2 (°)	2.17	1.35	1.21	1.14
RMSE_MIMU1 test3 (°)	3.07	0.78	0.66	0.61
RMSE_MIMU1 test4 (°)	3.54	0.82	0.74	0.67
RMSE_MIMU1 test5 (°)	3.22	0.71	0.69	0.62
RMSE_MIMU2 test1 (°)	3.19	1.34	1.2	1.1
RMSE_MIMU2 test2 (°)	1.36	0.96	0.9	0.8
RMSE_MIMU2 test3 (°)	2.33	0.79	0.7	0.59
RMSE_MIMU2 test4 (°)	2.97	1.17	1.04	0.86
RMSE_MIMU2 test5 (°)	4.19	1.54	1.12	0.87
Mean ± STD (°)	2.82 ± 0.78	1.10 ± 0.30	0.97 ± 0.26	0.86 ± 0.24

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
