# Peer review of "Magnetometer Calibration and Field Mapping through Thin Plate Splines"

_sensors, 2019, doi:10.3390/s19020280_

Reviewer 1 Report

Paper describes a method for to characterize and the 3D magnetic vector at every point of the measurement volume. Value of magnetic field is measured with magnetometer and algorithm for interpolation based on thin plate splines is used to interpolate the values between the measured points. Big advantage of the presented method compared to other methods is that the algorithm presented enables simultaneous estimation of the magnetometer calibration parameters.

Although the paper is generally well written, the results are not well presented. Clarity of the paper therefore suffers considerably.

Line 45, authors state: “However, as the distortions are temporally stable and spatially distinctive …” Does this means that in order for the method to be applicable the disturbances must be both temporally stable and spatially distinctive? If such, this should be clearly stated in the paper. Similarly, in line 130 authors state: deltaB is time-constant only in presence of motionless high permeability materials or static magnetic field generators in or near the measurement volume. Does this method works for time-constant magnetic fields or can they be time-varying.

Sentences in lines 140-149 are slightly confusing. Authors list from i) to iv) source of errors. Next sentence begins with “The latter includes…”, first one thinks that this relates to the last source of error listed in previous sentence, while it actually relates to magnetic deviation errors.

In lines 221-227 iterative method is presented. Lines from 1-3 are already explained in details in previous parts of the paper, but lines from 4-5 are not explained. How is sensitivity matrix calculated and how is the optimal correction calculated?

Class of tests 2 (lines 264-269): “the magnetometer explores different positions in the measurement volume (the same volume of the class of tests 1), but without exploring all the rotations.” Does this means that rotation is constant or the rotations actually change but in smaller amount?

Section 2.5.1 could be substantially shortened.

On what factors depends the selection of number and position of kernel points? Are they positioned randomly? I assume that in places with larger gradient of change of magnetic field there should be more points. Authors give explanation of this in the Discussion, but it would be helpful if some explanation would be also given in section methods where the method is explained.

Why is figure 2 showing only x component of the magnetic field. What does blue line shows and what does red line shows. Why are they presented with time on x axis. As far as I understood the filed depends on location and not on time. Shouldn’t the magnitude of magnetic field be different from zero (around 0.4 gauss). This comment also plies to figure 4.

The legend in figure 3 and description in text in lines 412-413 does not seem to correspond to each other. Legend says blue line is class1, while the text says it’s class 2.

Table 2 summarizes the errors in heading estimation. How do you know what is the real heading? How was the real heading measured?

Figures 5 shows the mean values; authors should also add two lines to show the STD values.

What was the actual number of frames? It is shown only in percentage.

Figure 5(a) and figure 6 show heading RMSE versus number of kernel points, but they are significantly different. Authors explain that “Finally, a comparison of the LRS magnetic field estimated in different acquisitions was performed. An example is reported in figure [6], where two different LRS magnetic field estimated in two different acquisitions, with different MIMUs, are compared:”. How were they compared, how did you actually arrive to presented values. Were the subtracted from each other. Authors should explain this part more in detail.

While the validity of the proposed method was indeed demonstrated using both simulated and real data, authors should also present numerical data how the method compares to similar methods. Is it better, is it more efficient, etc.

Minor observations:

In figures that have subfigures a) and b), a) and b) designations are set too much from the center of the subfigures. There is also a lot of empty space from lines 308 to 316.

Author Response

We would like to thank the reviewer for careful and thorough reading of this manuscript and for the thoughtful comments and constructive suggestions, which help to improve the quality of this manuscript. We attach the file with the point-by-point response

Reviewer 2 Report

The manuscript is well written, organized and presented with minor typographical, spelling and English use errors that should be corrected in the revised version.

The aim of the presented research is clearly defined against the current state of the art. 

However, certain clarifications concerning the methodology are deemed necessary.

In lines 141-143, the sources of errors are reported. The instrumentation error is composed of 4 components and the magnetic deviation is composed of two components. In Eq 2, two instrumentation error sources and one magnetic error source are represented by matrix W, two of the remaning error sources are represented by the bias matrix O and the sixth term is the additive noise which is not defined. Can you please elaborate on the nature and role of the additive noise?

What is the relationship of the above to the 0.0015G error reported in the actual sensor datasheet? 

In eq 4, m, P and R are inputs to the alogrithm . How are the W and O values, used to obtain the simulated m-values, generated in the simulated runs?  

Why is 100Hz used in the simulations while 120Hz are used for measured data?

How is the 5Hz BPF cutoff frequency chosen?

How do the x-axis results of Fig2 relate to the 3D results reported in the table below? How is the RMShE calculated for 3D?

No discussion is offered on the results of Fig 4 compared with Fig2.

line 484 reads "... testing many different parameter configurations": it is nto clear what is meant by this; can you calirfy or give examples? what parameters are you referring to?

Y-label axis is missing in Fig 2

l 75: correct "self - inducted" to "self-induced" 

Author Response

First, we would like to thank the reviewer for his/her valuable comments that for sure helped us in improving the paper. The comments are encouraging and the reviewer appear to appreciate this study and its results. We modified the manuscript according to reviewers’ specific comments and we hope that now it will be suitable for publication on Sensors. We attach the file containing the point-by-point response
